# Formation and Recognition of UV-Induced DNA Damage within Genome Complexity

**DOI:** 10.3390/ijms21186689

**Published:** 2020-09-12

**Authors:** Philippe Johann to Berens, Jean Molinier

**Affiliations:** Institut de Biologie Moléculaire des Plantes du CNRS, 12 rue du Général Zimmer, 67000 Strasbourg, France; philippe.johann-to-berens@etu.unistra.fr

**Keywords:** ultraviolet, photolesions, photodamage recognition, chromatin, photolyase, nucleotide excision repair, transcription coupled repair, global genome repair

## Abstract

Ultraviolet (UV) light is a natural genotoxic agent leading to the formation of photolesions endangering the genomic integrity and thereby the survival of living organisms. To prevent the mutagenetic effect of UV, several specific DNA repair mechanisms are mobilized to accurately maintain genome integrity at photodamaged sites within the complexity of genome structures. However, a fundamental gap remains to be filled in the identification and characterization of factors at the nexus of UV-induced DNA damage, DNA repair, and epigenetics. This review brings together the impact of the epigenomic context on the susceptibility of genomic regions to form photodamage and focuses on the mechanisms of photolesions recognition through the different DNA repair pathways.

## 1. Introduction

Solar radiation that reaches the surface of the Earth consists of 3 main spectra: ultraviolet (UV; 100–400 nm), visible light (400–700 nm), and infrared (IR; 700 nm to over 1 mm). Each of these ranges of wavelengths plays essential roles in providing light, heat, and energy, allowing the proper development of life. In addition to their beneficial impacts for living organisms, these types of irradiation can lead to deleterious effects, affecting cellular structures, interfering with biological processes, and damaging DNA.

The genotoxic effect of UV radiation from sunlight (UV-R; UVA: 320–400 nm and UVB: 280–320 nm) has been studied for a long time [1]. UV-induced DNA damages are formed between dipyrimidines, leading to DNA helix distortion and alterations of transcriptional programs [2]. To prevent such dramatic changes as well as mutations and genome rearrangements, specific DNA repair pathways are mobilized [2,3]. Depending on the organism and on the growth conditions, photodamages are processed by light-dependent (photoreactivation) and/or by light-independent repair pathways (i.e., nucleotide excision repair, NER), allowing an efficient maintenance of genome integrity [2,3].

These specific DNA damage repair pathways rely on a photodamage recognition step within the complexity of genomic regions displaying variations in accessibility (i.e., chromatin compaction). The emergence of sophisticated approaches to map the photolesions genome wide, to decipher (epi)genome shapes and protein occupancy at particular loci, allowed considering that DNA damage formation, together with the choice and the efficiency of the repair pathways, could likely be under the multifactorial influence of genome and epigenome organizations.

This review will describe the different types of UV-induced DNA lesions and will present the current knowledge in the putative interconnections existing between epigenetic marks and photodamage formation. In addition, the mode of action and the structural features of photolesions recognition factors, acting in the different DNA repair pathways, will be highlighted.

## 2. Impact of UV Radiation on DNA

The DNA, as support of the genetic information, is the target of UV-R. Indeed, nucleotides absorb UV-R, especially in the wavelength ranging from 100 to 280 nm (UV-C) and from 280 to 315 nm (UV-B) [4]. These short UV-R wavelengths can raise DNA bases to their highly reactive singlet or triplet states, which are prone to undergo different photochemical reactions. Interestingly, recent studies propose that DNA adopts “collective” excitation states, delocalized over at least two bases, when irradiated with UV-C/UV-B and to some extent with UV-A (315–400 nm; [5]). Three main types of DNA lesions are formed by these photochemical reactions and involve two successive pyrimidine bases (CC, TT, TC, and CT): cyclobutane pyrimidine dimers (CPDs), pyrimidine 6-4 pyrimidone photoproducts (6-4PPs), and their Dewar isomers (Figure 1a). CPDs can be detected nearly instantly after UV-R (1 ps) as a result of the formation of a stable ring structure between the C5 and C6 atoms of two adjacent pyrimidines [6]. 6-4PPs are formed in a slightly slower process (4 ms) involving the C4 of an oxetane or azetidine intermediates (at the 3′-end) and the C6 of the (5′-end) pyrimidine to build a stable noncyclic bond [7]. The quantum yields (Φ) ratio ΦCPD/Φ6-4 PP upon UV-R was estimated in a range of seven double-stranded DNA by in vitro and in vivo assays [8,9].

Upon absorption of UV-R, the 6-4PP photolesions can further evolve to their Dewar valence isomers in a fast (130 ps) electrocyclization reaction between the N3 and the C6 in the pyrimidine ring structure of the 3′ base of the 6-4PP (Figure 1a) [10]. The formation of these photolesions leads to weak base pairing reflected by changes in the helical DNA conformation [3]. Importantly, the DNA double helix distortion induced by the 6-4PP is much greater than that of the CPD [3]. In addition to the formation of direct photolesions, several indirect forms of DNA damage can occur by an interplay of photosensitization and oxidation reactions [11]. Through mechanisms of photosensitization, neighboring biomolecules excited by UV-R have the potential to directly or indirectly induce a chemical modification of the DNA. One example of such a photosensitization process is the triplet–triplet electron transfer (TTET) from UV-A excited benzophenone compounds to a nearly located thymidine to create a CPD photolesion [12]. Photosensitization mechanisms involving non-DNA chromophores were also described, by the side of enzymatic activation and bystander effect, as a predominant source of reactive oxygen species (ROS) upon UV irradiation [11]. In further reactions, ROS and especially hydroxyl radicals (OH•) can induce the oxidation of purine and pyrimidine bases and of the deoxyribose backbone of DNA [11]. The predominant indirect photodamage caused by the oxidative burst is the highly mutagenetic 8-hydroxyguanine (8-oxo-G; Figure 1a) and in a smaller extent, DNA single and double-strand breaks (SSB and DSB; (Figure 1a) [11,13]. Additionally, UV-induced ROS can indirectly lead to base alkylation and DNA–protein or DNA–DNA cross-linking [14,15]. The recognition of 8-oxo-G, SSB, and DSB [16,17,18,19] will not be reviewed in the following parts.

## 3. Influence of (epi)Genomic Features on Photolesions Formation

### 3.1. Dipyrimidines Composition

For decades, the susceptibility of the genome to form photolesions upon exposure to UV-R was thought to be quite homogenous, although the frequency and the genome-wide distribution of pyrimidine dimers (CC, TT, CT, and TC) may significantly differ among living organisms [20]. In human, CPDs and 6-4PPs are mostly formed between TT and TC, and with a lower frequency at CT and CC sequences [21]. In the model plant *Arabidopsis thaliana*, CPDs are predominantly formed between CT and TC and to a lower extent between TT and CC [22,23]. Given that dipyrimidines frequencies are quite similar between human and Arabidopsis [21,22,23], such differences in reactivity cannot be only explained by a strong bias in dinucleotides composition. Other factors such as sequence context and chromatin structure should also be considered as putative parameters influencing the formation of UV-induced DNA damage.

Indeed, the composition of neighboring nucleotide sequence of pyrimidine dimers was also shown to impact damage frequency, which is in agreement with the hypothesis of a putative “collective” excitation state [24]. All together, these observations highlight that photolesions formation likely differs within a genome but also between kingdoms. The availability of whole-genome sequencing data and photoproducts maps may provide an added value to better assess the underlying features of genome reactivity.

### 3.2. DNA Methylation

Recent studies hypothesized that the epigenomic context may influence the susceptibility of particular loci to form photolesions (Figure 1b). Indeed, in addition to the nucleotidic sequence, epigenetic marks (DNA methylation, histone post-translational modifications [PTMs]) may affect the UV-R-associated damaging processes. In eukaryotic cells, the DNA is mostly packaged into chromatin fiber. The smallest repeating units of these chromatin fibers are the nucleosomes, which are composed of 145 to 147 DNA base pairs (bp) wrapped around histone core proteins, which are separated from each other by a “linker DNA” of 20 to 100 bp (organism dependent) often complexed with the H1 linker histone [25]. The DNA binding to histones serves as a platform for PTMs to regulate, amongst others processes, gene expression, higher chromatin structure, and DNA repair [26]. Several in vitro assays started considering the role of cytosine methylation, relative nucleosome positioning, and protein binding in the damage formation. Indeed, the methylation of DNA at C5 of cytosine (5-mC), an important epigenetic mark regulating gene expression [27], was shown to increase by 80% the CPD quantum yield and to decrease by a factor 3 the 6-4PP quantum yield [8]. The higher sensitivity of 5-mC to form CPD could be due to the redshift of its absorption spectrum and its diminished amplitude of conformational motions in a DNA duplex [8,28].

Regarding the relatively high quantum yield of CPD compared to 6-4PP, 5-mC could be considered as an epigenetic mark favorizing photolesion formation (Figure 1b). Unlike animals, where DNA methylation is predominantly found in CG islands, DNA methylation occurs in 3 different contexts in plants: CG, CHG, and CHH (where H is A, C, or T) [29]. Therefore, it is tempting to speculate that plant genomes would be more prone to form photoproducts because of (i) their light-dependent lifestyle and (ii) the higher probability to find two consecutive pyrimidines in combination with a 5-mC (i.e., CTG or CCG in the CHG context). Such features have to be considered for deciphering genome responsiveness to UV-R (i.e., the formation of photolesions) and the complex interplays between DNA repair processes identified in plants [30].

The mapping of photolesions in a histone-loaded DNA context showed that upon UV-R, CPDs are formed around the core histone and in the linker DNA sequence, while 6-4PP is preferentially formed in the linker region of chromatin [31]. A deeper analysis at single-nucleotide resolution identified that CPD but not 6-4PPs occurred in a periodic pattern every 10.3 bp around the histone core [31] (Figure 1b). This periodicity reflects the conformational dynamics of DNA within chromatin. Every 10.3 bp, the DNA phosphate backbone is exposed away from the histone core, locally increasing its conformational motion potential and creating an energy “sink” [32,33]. Interestingly, due to the anisotropic bending preferences of the DNA, the more exposed sequences tend to be enriched in G and C, while the sequences close to the histone core tend to be enriched in A and T [34,35]. Interestingly, the ratio of CPD quantum yield in vivo versus naked DNA at sequences of strongly positioned nucleosomes appears to reach a maximum of around 1.2 for the exposed sequences and a minimum of around 0.9 for the sequence close to the nucleosome [31]. Thus, it is likely that DNA bound to nucleosomes is more prone to form CPDs than naked DNA. The in vivo impact of histone variants, histone PTMs, on DNA reactivity to form photolesions is a challenging area of research in the future.

### 3.3. Chromatin States

Taking into account the potential role of the above discussed genomic and epigenomic features in photoproducts formation, it could be expected that photolesions distribution may not occur randomly all over the genome in vivo.

According to the speculations of theoretical chemistry, in a biological system, a compacted DNA structure would be more reactive upon UV-R [5]. In the model plant *Arabidopsis thaliana*, this hypothesis seems to partially hold true. Indeed, CPD and 6-4PP mapping show significant enrichment in highly compacted heterochromatic regions [23].

In human, recent studies using HS (High Sensitivity)-damage-seq or ChIP (Chromatin Immuno-precipitation) assay followed by ELISA quantification did not reveal differences in CPD enrichment in specific chromatin states [21,36]. Nevertheless, CPD immune-staining assays show a non-homogeneous distribution of the photolesion upon acute UV-R, and several hotspots of photolesions have been identified [36,37]. How far the epigenetic context is involved in this hotspot formation, in humans, remain unclear. Similarly, in arabidopsis, genomic regions exhibiting heterochromatic features (high compaction, high DNA methylation levels) are more prone to form photoproducts, suggesting that particular epigenomic marks may contribute to such accurate reactivity [22,23].

Moreover, the impact of DNA binding proteins on photolesions formation was also taken into account. In vitro experiments showed altered reactivity of the binding sequences depending on the class of transcription factors (TF) [31]. In some cases, a subpart of the binding sequence even becomes a hotspot of photolesion, highlighting a putative role of DNA binding factors in genome damaging processes [38,39]. The UV reactivity of TF binding sites differs for each DNA binding protein and most probably depends on the conformational changes induced in the helical structure upon binding [39]. This hypothesis holds true in recent in vivo whole genome photolesion mapping assays, where the same TF binding sequence shows differential CPD enrichment at different loci depending on the secondary binding proteins [31]. In conclusion, protein binding is neither strictly correlated nor anticorrelated with a higher reactivity of DNA upon UV-R in vivo.

Considering these several lines of evidence supporting the idea that genomic and epigenomic contexts likely influence the formation of photolesions, the DNA repair pathways may have specifically evolved to efficiently recognize such damage in the complexity of the different chromatin landscapes.

## 4. Photolesion Repair Pathways

Two main strategies exist to repair UV-induced DNA lesions. A light-dependent process (referred as “light repair”) that reverts photodamage using particular wavelengths and a light-independent process (referred as “dark repair”) that excises the UV-damaged region followed by de novo synthesis of an intact DNA strand. Although most of the living organisms possess both pathways, the light repair pathway is predominantly used [40]. Importantly, growth conditions (i.e., full light versus shadow), tissue specificities (i.e., roots versus leaves), and the transcriptional level of particular genomic regions (i.e., euchromatin versus heterochromatin) are some examples of parameters that could determine the predominant use of one or the other pathway. For each of these pathways, specific factors/complexes recognize the photolesions and trigger the repair process.

### 4.1. Light Repair

An essential repair pathway of photon-induced damage is the direct repair (DR) pathway, which, interestingly, depends on photon-triggered enzymes called photolyases (PLs) (Figure 2a) [41,42]. Photolyases perform the repair of photolesions by reverting the damage [42]. In other words, this repair pathway does not rely on de novo DNA synthesis. According to phylogenetic analyses, it was proposed that 3.8 billion years ago, all living organisms possessed photolyase-like genes, making DR the oldest known DNA repair mechanism [43,44]. DNA photolyases genes evolved in all branches of life, including eukaryotes [44]. However, PLs are not found in placental mammals but exist in marsupials [44,45,46]. Despite their high sequence and structure similarities, photolyases with conserved DNA repair activity are distinct from cryptochromes (CRYs), which gained new functions as a light receptor involved in the regulation of gene expression or phototaxis [44,47]. PLs can further be classified as CPD- or 6-4-photolyases, according to their exclusive substrate specificity for CPDs or 6-4PPs, respectively [48]. Extensive studies in several model organisms such as *E. coli* [49,50], *S. cerevisiae* [51,52], *D. melanogaster* [53,54], and *A. thaliana* [55,56,57] allowed deciphering the specificities and the modes of action of several PLs. Surprisingly, a bifunctional photolyase, with a CPD and 6-4PP substrate recognition and repair activities, was recently identified [58].

PLs are structurally composed of an N-terminal dinucleotide binding domain and a C-terminal binding domain for the catalytic cofactor: flavin adenine dinucleotide (FAD) [42,47]. Besides FAD, many PLs also bind additional chromophore such as methenyltetrahydrofolate (MTHF) or 8-hydroxy-7,8-didemethyl-5-deazariboflavin (8-HDF) [42,47]. The electrostatic surface potential map shows an accumulation of positively loaded residues flanking a cavity localized in the vicinity of the flavin cofactor [59,60]. These positively charged residues bind the negatively loaded phosphate backbone of the DNA helix, whereas the hydrophobic cavity specifically binds the pyrimidine dimer [59,61]. Localization of the pyrimidine dimer in the enzymatic active binding pocket depends on a helical out-flipping of the damaged nucleotides (Figure 3a) [59,60,61,62]. The most recent study of CPD photolyase substrate binding kinetics suggests that conversely to other DNA repair proteins, CPD recognition does not rely on one-dimensional sliding or hopping along the DNA, but on the three-dimensional search for an extrahelical out-flipped photolesion [63,64]. Once out-flipped, the intrahelical bubble is stabilized by a bubble-intruding region (BIR) in Class II photolyases [65,66] and by a conserved Arg421 in 6-4PP photolyases (Figure 3a) [61,62]. In Class I photolyases, the structure is most probably stabilized by another type of interaction that remains to be further characterized [59,67].

The above described binding structures were always determined in a nucleosome-free environment [59,61]. Given that the binding of photolyases induces a local DNA bending [59], the chromatin environment might be recalcitrant to such conformational change and hence would inhibit the recognition process [68]. Indeed, in yeast, in vivo photolyase-mediated photolesion repair is slowed down in nucleosome-bound regions [68]. However, photolesions located in the core regions of the nucleosome can also be repaired by PLs, but this process needs more time, arguing in favor of a chromatin remodeling mechanism [68]. Importantly, no shreds of evidence for a photolyase-specific chromatin remodeling mechanism have been described so far. Upon stable binding to the CPD or 6-4PP photolesions, the photolyase performs the “direct repair” reaction (Figure 2a). For this purpose, FAD and additional photo-antenna molecules collect energy through the absorption of a blue light spectrum photon [42]. The energy transfer generates excited FADH^−^• [42]. In the case of CPD photolyases, FADH^−^• donates an electron to the CPD to catalyze the reversal repair reaction by cleaving the C5-C5 and C6-C-6 bonds of the cyclobutane ring [42,48]. The repair reaction of 6-4 photolyases also uses FADH^−^• as an electron and proton donor to generate a transient oxetan-type residue followed by C6-C4 bond splitting [42,69]. In both cases, the result is the restoration of the native DNA sequence and photolyase release in a DNA synthesis-independent manner [42].

### 4.2. Dark Repair

The dark repair pathway, also called NER, promotes the repair of photolesions in a light-independent manner via two sub-pathways: transcription coupled repair (TCR) and global genome repair (GGR) processing photodamage along actively transcribed DNA strands or throughout the genome, respectively. NER is a DNA synthesis-dependent repair pathway. Thus, it implies that, in addition to the nucleotidic sequence, the epigenomic landscape (i.e., DNA methylation) must be accurately re-established.

#### 4.2.1. Transcription Coupled Repair

The first experiments providing evidence for the existence of a TCR pathway in eukaryotes was performed on Chinese hamster ovary deficient in global genome repair [70]. The authors showed that CPD repair was more efficient in transcribed genomic regions compared to the transcriptionally silent upstream sequences [70]. The TCR damage recognition step was shown to rely on the stalling of RNA Polymerase II (RNA Pol II) [71]. As a consequence, TCR predominantly repair lesions on the transcribed DNA strand [71,72]. The RNA Pol II translocates along the DNA template strand, synthesizing the complementary RNA molecule. Gaps, breaks, and modified nucleotides can lead to stalling and arresting of the polymerase (Figure 2b) [73]. This stalling, identified to be the recognition step [71], mainly depends on the two highly conserved critical residues (R1386 and H1387) in the switch1 region of Pol II, which is described as a sensor of structural barriers in the minor groove of the DNA helix upstream of the polymerase (Figure 3b) [74]. However, RNA polymerase stalling can also be induced, in the absence of DNA damages, by extra-stable chromatin structures, a weak affinity between DNA and nascent RNA, or secondary structure in the nascent RNA [73].

Interestingly, a recent study investigated the role of early TCR factors in a potential differentiation mechanism, which can help to overcome some type of obstacles or lead to the recruitment of the damage repair machinery [75]. Considering these observations, the decisive recognition step only occurs by an interplay between RNA Pol II and the SWI2/SNF2 (SWitch 2/Sucrose Non fermenTable 2) protein Rad26 in *Saccharomyces cerevisiae,* which is the homolog of the human CSB (Cockayne Syndrome protein B) and the arabidopsis CHR8 proteins [75]. Mutations in the CSB gene result in a rare genetic disease called “Cockayne syndrome” [76]. In the proposed model, Rad26/CSB/CHR8 binds stalling RNA Pol II between the clamp (Rpb2 side) and stalk (Rpb4/7) regions, and it promotes its forward translocation, increasing the bypass efficiency at minor barriers [75,77]. While base alkylation [77], abasic sites [78], and 8-oxo-G [79] can be bypassed by the RNA Pol II, photolesions induce stalling and arrest [80,81]. In the example of (T<>T) CPD, the stalling occurs by the stacking above the bridge helix of Pol II (Figure 3b), slow incorporation of an A in front of the first T involved in the dimer, and an even slower misincorporation of a U in front of the second T [81,82]. This misincorporation finally leads to the arrest of transcription [81,82].

The persistent binding of Rad26/CSB/CHR8 to the arrested RNA Pol II signals the sequential recruitment of the NER machinery to complete the recognition step. The CSA (Cockayne Syndrome protein A)–Cullin 4 E3 ubiquitin ligase complex is recruited to ubiquitinate CSB and the Pol II subunit RPB_1_ at position K1268 [75,83,84]. The stability of CSB seems to be regulated by a complex interplay between SUMOylation (Small Ubiquitin-like MOdifier) and ubiquitination homeostasis, opposing the CSA–Cullin 4 E3 ubiquitin ligase complex and the ubiquitin C-terminal hydrolase 7 (USP7) [85,86,87]. In parallel, the monoubiquitination of UVSSA (UV-sensitive syndrome (UV^S^S) A), at position K414, triggers TFIIH (Transcription Factor II H) recruitment, leading to DNA unwinding (Figure 2b,e) [83,88]. TFIIH binding requires USP7 to leave the complex, enabling CSB polyubiquitination and release. At this point, TFIIH is proposed to forward-translocate on the DNA using its 5′–3′ XPD helicase promoting the Pol II backtracking, in order to efficiently access the damage site [89,90,91]. Importantly, RPB_1_ polyubiquitination triggers RNA Pol II targeting to the 26S proteasome for degradation only when TCR is not functional in order to remove the arrested transcription complex from the DNA template [84,92]. After TFIIH binding and prior repair, a further step of validation of photodamage recognition is performed. This mechanism will be discussed in Section 4.2.3.

The core TCR process seems to be globally conserved in eukaryotes [93,94]. Even in drosophila lacking for CSB, CSA, and UVSSA homologs, a recent study revealed the existence of a TCR-like process [95]. Interestingly, in bacteria, the coupling factor Mfd (Mutation Frequency Decline) [96,97,98], which autonomously translocates on DNA, patrols for stalling polymerase [99]. Indeed, the binding of Mfd on a stalling RNA polymerase promotes its translocation [100]. If the arrest persists, because of severe obstacles, Mfd induces displacement of the RNA Pol and recruitment of UvrA, UvrB, and UvrC for NER. [97,98]. Alternatively, another TCR recruiting mechanism, independent of Mfd, was proposed to occur after RNA Pol II backtracking promoted by UvrD [101,102].

The RNA Pol II is not the only RNA polymerase stalling at UV-induced photolesions. A recent study showed that RNA Pol I stalled and arrested even earlier than the Pol II when encountering CPD [103]. In addition, RNA Pol I was also shown to form a complex with CSB [104] and to interact with TFIIH [105]. Besides, TCR was also observed in rDNA regions [106]. More recently, the analysis of the Pol I behavior upon UV irradiation revealed a considerable backtracking capacity but a low dissociation rate [107]. Altogether, these facts argue in favor of an alternative TCR pathway involving Pol I as a damage recognition platform that is so far poorly understood. Therefore, it may be of interest to reconsider the RNA Pol III involved in tDNA and in 5S rDNA transcription [108] and the plant-specific RNA Pol IV/Pol V [109], which are evolutionarily related to the RNA Pol II [110] as putative key players of the DNA repair machinery. Both RNA Pol IV/Pol V predominantly act in genomic regions containing high DNA methylation and compaction levels [29], suggesting that a non-canonical TCR process may exist in plants or that complex photodamage repair mechanisms may have evolved.

In brief, the TCR pathway acting in transcribed genomic regions displays an efficient recognition mechanism of UV-induced pyrimidine dimers. This process depends on the stalling of the RNA Pol II and the complexation with CSB at the damaged site (Figure 2b). This implies that transcriptional activation directly promotes the control of genome integrity. Consequently, epigenetic features impacting transcription initiation and elongation may also indirectly regulate genome surveillance pathways.

#### 4.2.2. Global Genome Repair

In addition to TCR, the global genome repair pathway (GGR) acts in poorly transcribed/untranscribed genomic regions to efficiently repair photolesions. In this NER sub-pathway, the damage recognition is performed independently of RNA Pol II [3]. In GGR, the central actor is the *Xeroderma Pigmentosum* complementation group C (XPC)–RAD23 protein complex (hereafter called the XPC complex). This complex was identified as the initiator of GGR because of the ability of XPC to bind DNA lesions (Figure 2d) [114]. The XPC–RAD23 complex is well conserved in eukaryotes: RAD4–Rad23 in *Saccharomyces cerevisiae* [115], XPC–RAD23B in human [116], and XPC–RAD23 in *Arabidopsis* [117,118]. RAD23 binds RAD4 through its R4BD domain (Figure 3d) [112], thereby regulating RAD4 stabilization [119] and promoting lesion recognition activity [120]. Additionally, the XPC complex is stabilized at damage sites when associated with Cdc31 and Rad33 in yeast [121,122]/Centrin2 in human [123,124] and AtCentrin2 and CML19 [125,126] in Arabidopsis [121,122,123,124,125,126].

The first crystal structure of the RAD4–RAD23 complex binding a CPD containing DNA helix confirms the underlying recognition process previously described for the human XPC by Maillard et al. [127,128]. RAD4 contains TGD (Transglutaminase homology domain), BHD1 (ß-hairpin domain 1), BHD2, and BHD3 domains (Figure 3d). TGD and BHD1 regions bind to 11 bp of undamaged, double-stranded DNA (Figure 3d). Simultaneously, the BHD2 and BHD3 domains bind to a 4 bp DNA lesion site by insertion of a β-hairpin of the BHD3 in the helix (Figure 3d) and a groove of BHD2–BHD3 interacting with the backbone of the undamaged strand (Figure 3d) [127]. This structure forces the damaged dimers to flip out of the helix structure, leaving them accessible (Figure 3d) [127].

To stably adopt the bound conformation, the XPC complex needs to overcome a consequent energy barrier, which was described as a primary regulator for the recognition specificity [129,130]. Indeed, DNA damages induce structural changes of the DNA helix structure and weak base pairing, leading to a decrease of the energy barrier that the recognition complex needs to overcome for efficient binding [131]. In other words, XPC may patrol along the DNA until encountering a disturbed helical structure with weak base pairing, allowing XPC to stably bind the lesion site [128,129,130,131,132]. This process likely explains how the XPC complex detects several different types DNA damages and why the identification of particular lesions is more efficient. For example, 6-4PP lesions causes a massive thermodynamic destabilization of the helical structure [133,134]. As a consequence, 6-4PP recognition by the XPC complex is preferred compared to CPD [135].

Additionally, the recognition efficiency is limited by the residence time of the XPC complex at lesion site [129]. By single-molecule tracking, XPC complex was shown in three different motions: (i) sliding on DNA, most probably scanning for damage site; (ii) in a constrained motion, approximatively 2 kb around the DNA lesion, and (iii) in the non-motile complexes [132]. As in TCR with the arrested RNA Pol II and CSB, the persistent binding of the XPC complex at the damage site recruits TFIIH and further NER factors (described hereafter in Section 4.2.3), ending the recognition step of the GGR.

Cell-free systems have significantly contributed to improving the understanding of the underlying mechanism of XPC photodamage recognition [127,129,132]. However, most of these systems used DNA substrates of relatively small sizes without nucleosomes and hence did not consider the substantial complexity of the recognition step in vivo. Unlike TCR, the GGR also acts in transcriptionally repressed regions with a high nucleosome density [136]. To efficiently fulfill its role in damage recognition in the context of chromatin, the XPC complex is assisted by damage pre-recognition and chromatin remodeling mechanisms.

The pre-recognition mechanisms involve the UV–DDB complex composed of DDB1 and DDB2 (DNA damage binding proteins 1 and 2), which are also known in human as p127 and p48, respectively (Figure 2c). Mutations in this complex lead to repair deficiency and UV sensitivity [137,138,139]. The UV–DDB complex enables a recognition of 6-4PP, CPD, mismatches, and apurinic/apyrimidinic sites in vivo [111,140,141]. The recognition of different damage is based on a mechanism of helical structure stability verification, resembling the scanning mechanism previously described for the XPC complex [111,142]. The human DDB2 contains a helix–loop–helix domain (residues 101 to 136) and a 7-bladed WD40 β-propeller domain (residues 137 to 455). The DNA binding of the UV–DDB complex is exclusively performed by the β-propeller. DDB2 binds 7 bp DNA by charge-stabilizing hydrogen bonds to the phosphodiester backbone [111,142]. This binding depends, among other residues, on a well-conserved Lysine (Lys244 in human) [111,142,143]. The point mutation of Lys244 causes DDB2 loss of function [143]. As described for the XPC complex, DDB2 can trigger a helical out-flipping of the lesion in the condition of helical structure distortion and weak base pairing (Figure 3c) [111,142]. This strand separation depends on the insertion of a 3 bp residue in DDB2 hairpin Phe334, Gln335, and His336 (Figure 3c) [111,142]. Unlike the XPC complex, which only interacts with the undamaged strand, DDB2 binds the displaced DNA lesion in a shallow pocket [111,142]. However, the shape and the composition of the pocket do not provide lesion binding specificity but limit the size and the chemical nature of the lesion that can be recognized [111,142]. This recognition mechanism holds true if the DNA is wrapped around a nucleosome, and a register shifting seems to be sufficient to allow DDB2 to stably bind the lesion (Figure 3c) [111]. In that way, DDB2 can act as UV-induced DNA damage pre-recognition platform, even in the context of dense chromatin [111].

The stable DDB2 binding to the lesion site activates the Cullin4 E3 ubiquitin ligase complex, which interacts with UV–DDB through the three β-propeller domains of DDB1 [144], leading to the ubiquitination of DDB2, XPC, and nearby histones [145,146]. Additionally, the UV–DDB complex was shown to recruit several chromatin remodelers [147,148,149] and histone methyltransferases: ASH1L [150] DOT1L [151], and NSD2 [152], suggesting that histone PTMs may also play a critical role in this recognition process [153].

The exact timing of interactions and the underlying interplays are only partially understood. ASH1L was shown to ensure the DNA lesion handoff between a DDB2 pre-recognition complex and the XPC recognition complex through H3K4 tri-methylation [150]. Additionally, the DDB2 dynamics are tightly regulated by PTMs, modulating its stability and retention time on chromatin [154]. Indeed, after stable damage recognition, DDB2 can be ubiquitinated [154], SUMOylated by PIASy (Protein inhibitor of activated STAT Y) [155,156], and poly-ADP-ribosylated by PARP1 (Poly ADP-Ribose Polymerase) [149]. The poly-ADP-ribosylation of DDB2 increases its retention time on chromatin [149]. In contrast, the ubiquitination leads to the proteasomal degradation of DDB2 to complete the pre-recognition step [154,157].

Although the exact function of SUMOylation is as yet undetermined, a recent study highlighted the role of SUMOylation in DNA–protein cross-link (DPC) labeling and clearance in higher eukaryotes [158]. Considering the increased probability of DNA–protein cross-linking upon UV exposure [15] and the SUMOylation of several other NER proteins [159] such as XPC [160], SUMOylation may serve as priming mark for DPC surveillance. This hypothesis is reinforced by the recent evidence that DDB2 can recognize another type of UV-induced DNA lesion, 8-oxo-G [161], which shows a considerable DPC reactivity [14,15]. Alternatively, SUMOylation may feed the recently described SUMO-Targeted Ubiquitin Ligases (STUbL) process to promote protein ubiquitination [162].

Histone ubiquitination and chromatin remodelers likely play a promoting role for the efficiency of the following NER steps, as shown for the CHD1 chromatin remodeler, ensuring the XPC to TFIIH handover of UV photolesions bound to nucleosomes [163].

However, DDB2 was shown to spatially and temporally regulate XPC recruitment and thereby NER (Figure 2c,d) [164]. In the nucleosome-bound sequence, a close handover may occur through the interaction between DDB2 and the BHD1 domain of XPC at the damage site [164]. By this mechanism, XPC may access compact chromatin and stick the damaged site for a later repair [164]. Furthermore, DDB2 can also induce chromatin relaxation, which may prefer a contactless handover to XPC [165]. One working hypothesis would be that photolyases could also take advantage of this relaxed chromatin microenvironment formed by DDB2 to access photolesion in heterochromatic structures.

In *Saccharomyces cerevisiae*, Rad7 and Rad16 functionally substitute the UV–DDB2 complex by an ATP-dependent UV-damage sensor [166]. The Rad7–Rad16 complex interacts with elc1 and Cullin3 to form a Cullin-based E3 ubiquitin ligase necessary for Rad4 ubiquitination in response to UV radiation [167,168]. The exact mechanism and sequence context of Rad7–Rad16 lesions binding remains unclear.

Interestingly, Rad7–Rad16 and UV–DDB are both conserved in the model plant *Arabidopsis thaliana* [169]. The UV–DDB pathway was already extensively characterized [157]. Importantly, in addition to the canonical DDB2 recognition pathway, *Arabidopsis thaliana* evolved a small RNA-mediated photolesion detection mechanism [22]. The model proposes that upon UV exposure, small RNA, with sequence complementarity to DNA damaged sequences, accumulate. Upon UV exposure, these 21-nt UV-induced small RNAs (uviRNA) are loaded into ARGONAUTE 1 (AGO1) and form a complex with DDB2 to be targeted at damaged site, leading to efficient photolesions recognition [22].

To summarize, GGR is composed of an essential pre-recognition step performed by the UV–DDB/Rad7-16 complex, which enables, among other processes, the recognition of 6-4PP and CPD in DNA bonded to nucleosomes [111,166]. This pre-recognition step induces histone methylation [153] and chromatin remodeling [147,148,149] to promote the recruitment of the XPC complex for the central damage recognition step of the GGR pathway [150,164]. Alternatively, the damage handover between DDB2 and XPC may occur in a transient interaction at damage sites in nucleosome-rich regions [164]. DNA binding of the UV–DDB complex also activates the Cullin4 ubiquitin E3 ligase complex, which, ubiquitinated DDB2 and XPC to coordinate the end of the pre-recognition and recognition steps [145,154]. Once bound to the damage site, the XPC complex, in an interplay with the chromatin remodeler CHD, recruits TFIIH for the validation step of the NER recognition process [163].

#### 4.2.3. Validation of NER Recognition Steps

Both TCR and GGR recognition steps end up by recruiting the TFIIH complex (Figure 2e), which is well conserved among eukaryotes [170,171,172,173]. In human, TFIIH is composed of a core complex with the 2 DNA helicases XPB and XPD as well as p62, p52, p44, p34, and p8 [174]. The additional CDK-activating kinase subcomplex is formed by CDK7, cyclin H, and MAT1 subunits, which are required for transcription initiation but not for DNA repair [175]. In the context of NER, TFIIH interacts with XPA and XPG (*Xeroderma Pigmentosum* complementation group G) instead of MAT1, inducing a conformational change [113]. XPA clamps the TFIIH complex to DNA, whereas the endonuclease XPG competes with MAT1 [113]. XPA binds XPB with its extended helix, forming a tunnel for the DNA helix and promoting the translocation activity of XPB [113]. Simultaneously, XPA intercalating hairpin interacts with XPD at the 5′-edge of the DNA repair bubble, promoting strand separation and the 5 —> 3′ helicase activity of XPD [113].

XPD performs the final DNA lesion recognition step of NER, which is also called the “validation step”. The mechanism by which the DNA strand is loaded into XPD was proposed to depend on the interaction between the HD2 domain of XPD and the ssDNA of the repair bubble [176]. This contact may subsequently initiate a transient opening of the interface between the Arch and the iron–sulfur cluster (FeS) domains to slip the DNA strand inside a cavity between the ATPase lobe1, the FeS cluster domain, and the Arch domain (Figure 3e) [176]. The damaged DNA strand is actively translocated through this cavity, enabling the proofreading DNA bases and the recognition of abnormal structures such as CPDs [113,176,177,178]. Indeed, the amino acids Y192 and R196 of the FeS domain, stabilizing the sugar–phosphate backbone, were shown to be essential for XPD retention at a bulky DNA lesion to (Figure 3e) [178]. The retention at the damaged site was proposed to depend on the lack of DNA-mediated charge transfer (CT) [179,180]. According to this hypothesis, electrons can be transferred between two FeS cluster proteins through the undamaged DNA duplex [179]. In the case of XPD, the amino acids R112 and C134, shaping a bridge between DNA and the FeS cluster, may allow the electron transfer, promoting the displacement of XPD (Figure 3e) [179]. In the presence of DNA damage, the CT through the DNA is altered, and XPD is stabilized, thereby labeling the damage site [179].

This last recognition step is followed by the recruitment of the endonucleases ERCC1–XPF and XPG in 5′ and 3′ of XPD, respectively [175]. These endonucleases perform a dual incision releasing a 30-nt DNA fragment containing the photolesion [175]. At the same time, the non-damaged strand is protected by the Replication Protein A (RPA) [175,181]. This excision step is followed by de novo DNA synthesis and nick ligation to restore the original DNA sequence (Figure 2) [175,182].

## 5. Conclusions and Perspectives

This review brought together several lines of evidence highlighting the existence of potential connections between the (epi)genomic landscapes, photolesions formation, and processing.

Indeed, genomic and epigenomic contexts (nucleotides composition, DNA methylation, nucleosome binding) lead to a differential reactivity of loci to form photolesions. A higher chromatin structure, determined by the epigenetic landscape, pins down the frame for recognition and repair specificities. In addition to the motif/amino acid-based recognition mechanisms of the core protein complexes, sequential handover between recognition and repair factors is partially mediated by chromatin remodelers and epigenome writers [150,153,183].

Given that the recognition step promotes histone eviction/sliding and also de novo DNA synthesis, it becomes evident that the accurate re-establishment of the epigenomic landscape is a part of the genome maintenance process. However, epigenome changes at damaged sites have been reported [23,184,185]. Such alterations may modulate the transcriptional programs, redirecting the choice of the photodamage repair pathway to be mobilized. Hence, it is likely that variabilities of DNA methylation, base composition, UV damage formation, recognition, and repair contribute to genome evolution [186]. This extended view on photodamage recognition mechanisms highlights the importance of future works to study the chromatin landscape at damaged loci upon UV exposure and repair, and it paves the way toward new concepts regarding the evolution of eukaryotic (epi)genomes.

## Figures and Tables

**Figure 1 ijms-21-06689-f001:**
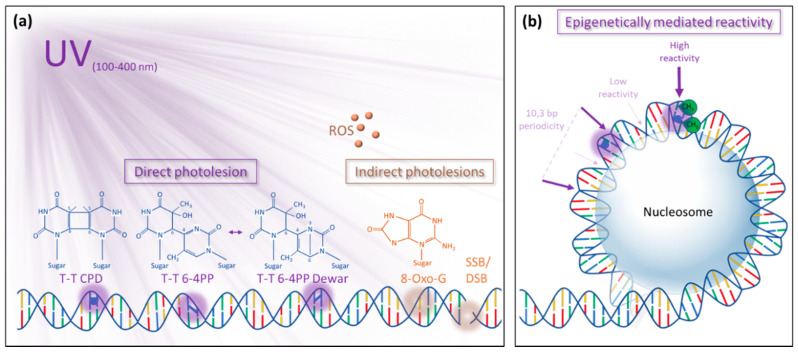
Photolesions and genome reactivity. (**a**) Schematic representation of the chemical structure of the most frequent direct and indirect photolesions induced upon UV exposure. From left to right: Cyclobutane pyrimidine dimer (example within two thymines T-T CPD), 6-4 photoproduct (example within two thymines T–T 6-4PP), Dewar valence isomer of the T-T 6-4PP (T-T 6-4PP Dewar), 8-oxo-7,8-dihydroguanine (8-oxo-G) indirectly induced by reactive oxygen species (ROS), single or double-strand breaks (SSB/DSB). (**b**) Schematic representation of the epigenetically mediated context reactivity to form photodamage upon UV radiation. Dark violet and light violet arrows signify a high or low sequence reactivity, respectively, compared to “naked DNA”. Methylated cytosines are labelled with a green CH_3_ group. CPD: cyclobutane pyrimidine dimers, 6-4PP: 6-4 pyrimidone photoproducts.

**Figure 2 ijms-21-06689-f002:**
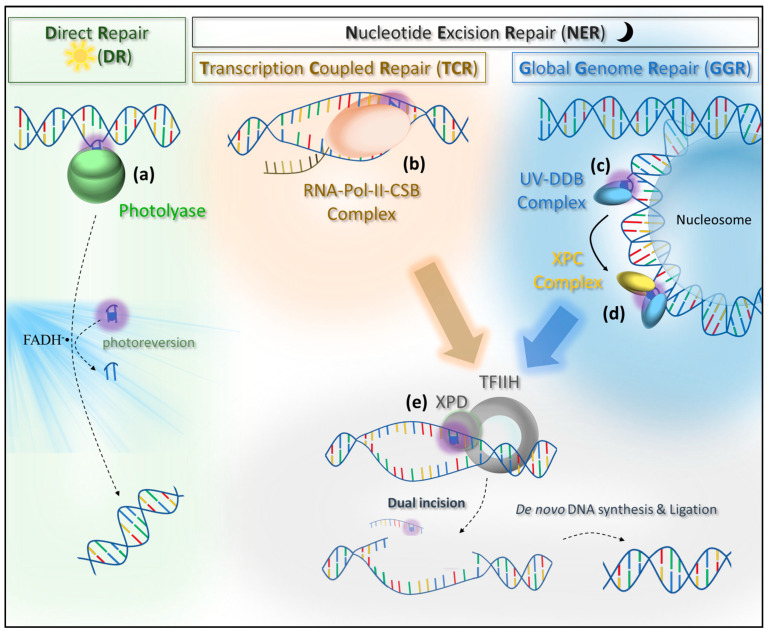
Photolesion recognition and repair pathways. (**a**) The direct repair pathway (light repair) relies on specific photolyases, which either recognize CPD or 6-4PP. The photolyase interne flavin adenine dinucleotide (FAD) cofactor, excited by blue light, catalyzes photoreversion to restore the initial undamaged sequence. (**b**) The transcription coupled repair (TCR) pathway (dark repair) is specific to transcribed genomic regions and depends on the RNA Pol II–CSB (RNA Polymerase II-Cockayne Syndrome protein B) complex for the recognition step. RNA Pol II stalls and arrests at the damage site. (**c**) The global genome repair (GGR) pathway (dark repair) primarily recognizes the photolesion by the damage sensor complex UV–DDB (DNA damage binding protein), which is able to scan DNA in compacted chromatin. (**d**) Once bound to the damage, UV–DDB recruits the Rad4/XPC (*Xeroderma Pigmentosum* complementation group C) complex for a second recognition step. The stalled RNA Pol II–CSB and The Rad4/XPC complex recruit the TFIIH (Transcription Factor II H) protein complex. (**e**) XPD (*Xeroderma Pigmentosum* complementation group D) proceeds to a damage validation step. Upon this final recognition step, the damaged DNA region is excised by a dual incision process, and the gap is filled by de novo DNA synthesis and nick ligation.

**Figure 3 ijms-21-06689-f003:**
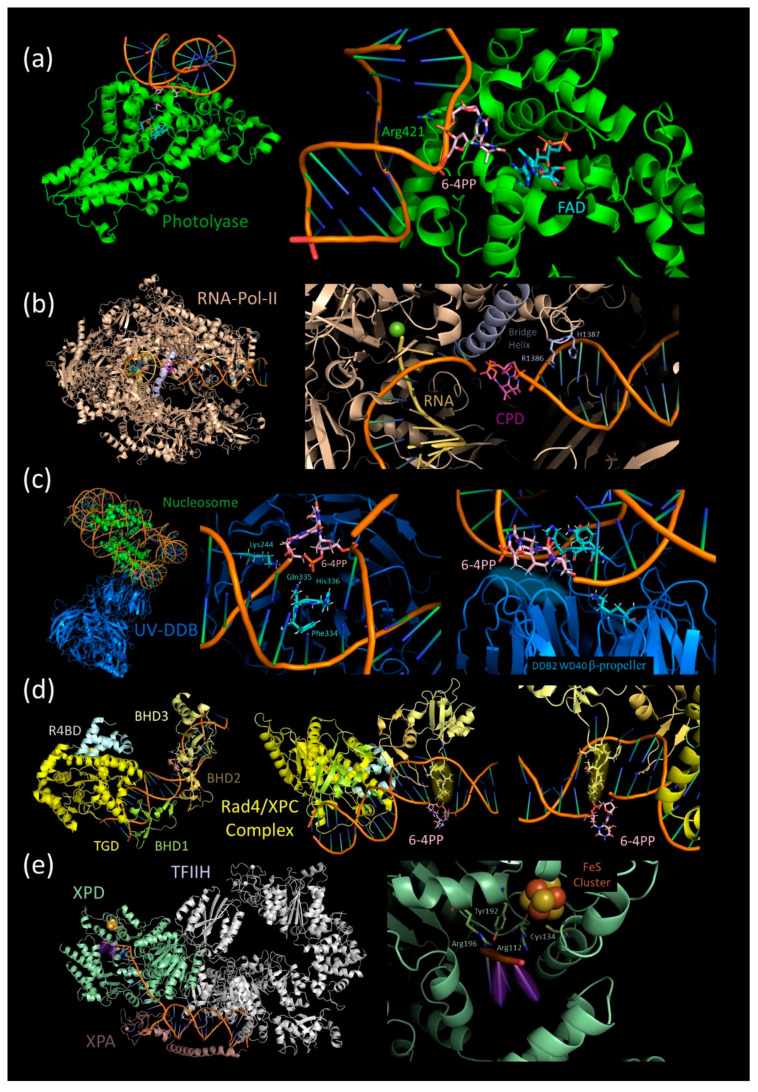
Structural views of the main photolesions recognition factors. (**a**) Left panel: in silico modeling of the *Drosophila melanogaster* (6-4) photolyase bound to double-stranded (ds) DNA with a “out-flipped” T-T 6-4PP. Figure is based on the PDB structure 3CVU [62]. The photolyase and these residue Arg421 are colored in green, the catalytic FAD ligand is colored in cyan blue, and the 6-4PP is in light pink. Right panel: Zoomed-in detailed view of the photolyase–lesion interaction. (**b**) Left panel: in silico modeling of a *Saccharomyces cerevisiae* RNA polymerase II elongation complex arrested at a CPD lesion. Figure based on the PDB structure 6O6C [75]. The RNA Pol II is colored in salmon, the Pol II bridge helix and the residues R1386 and H1387 are colored in gray-blue, the nascent RNA is in gold, and the CPD is in violet. Right panel: Zoomed-in detailed view of the RNA Pol II–lesion interaction. (**c**) Left panel: In silico modeling of the *Homo sapiens* UV–DDB complex bound to a “out-flipped” 6-4PP in double-stranded DNA (dsDNA) wrapped around a nucleosome. Figure is based on the PDB structure 6R8Y [111]. The UV–DDB complex is colored in blue, the DDB2 residues Lys244, Phe34, Gln335, and His336 are in cyan, the nucleosome is in green, and the 6-4PP is in light pink. The photolesion binding pocket is highlighted with a blue hallow. Right panel: Zoomed-in detailed view of the DDB2–lesion interaction. (**d**) Left panel: In silico modeling of a *Saccharomyces cerevisiae* Rad4–Rad23 complex (XPC complex homologue) bound to a 6-4PP photoproduct. The figure is based on the PDB structure 6CFI [112]. The Rad4 TGD (Transglutaminase homology domain) is colored in yellow, the BHD1 (ß-hairpin domain 1) is in lime-yellow, the BHD2 domain is in gold, the BHD3 domain is in pale-yellow, the Rad23 R4BD domain is in white, and the 6-4PP is in light pink. Right panel: Zoomed-in detailed view of the *Xeroderma Pigmentosum* complementation group C (XPC)–lesion interaction. The helix insertion hairpin from the BHD3 domain is highlighted with a yellow hallow. (**e**) Left panel: In silico modeling of the *Homo sapiens* core TFIIH–XPA–DNA complex without photolesion. Figure is based on the PDB structure 6RO4 [113]. The TFIIH is colored in white, the XPA (*Xeroderma Pigmentosum* complementation group A) protein is in salmon, the FeS cluster is in yellow and orange, and the XPD protein and its Arg112, Cys134, Tyr192, and Arg196 residues are in pale green. Right panel: Zoomed-in detailed view of the XPD–lesion interaction. The theoretical localization of the photolesion during the recognition step is highlighted with a violet hallow. All figures were created using PyMOL (The PyMOL Molecular Graphics System, Version 2.0 Schrödinger, LLC.).

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
