# Peer review of "Formation and Recognition of UV-Induced DNA Damage within Genome Complexity"

_ijms, 2020, doi:10.3390/ijms21186689_

Round 1
Reviewer 1 Report
The authors provide a useful update on the research related to the formation and recognition of UV lesions across all kingdoms of life.
Unfortunately, my version of the submitted manuscript lacks the last 18 references such that I would need to see the complete manuscript to finalize the review. In the meantime, I hope that the following comments can be included into the draft.
- Lines 93-96: What is the presumed reason for the intriguing difference in the sequence preference of CPD formation between human and plant cells? Is there any hypothesis?
- Line 265: What ist the postulated role of PolII ubiquitination during the TCR process?
- Lines 265-268: The authors describe the push-forward action of CSB but not the possible backtracking function of TFIIH as described for example in the review by Lans et al. (2019) Nat. Rev. Mol. Cell. Biol. 20, 766-84.
- Line 337: The unique mechanism of substrate recognition by XPC protein (relying on the detection of a single-stranded configuration in the undamaged complementary sequence of the double helix) has first been reported by Maillard et al. (2007) PLoS Biol. 5:e79. This reviewer believes that credit should be given to this pioneering study that appeared before the description of a crystal structure of the yeast homolog.
- Lines 348-349: The mechanism by which XPC protein dynamically searches for DNA damage has been described in the context of living human cells by Camenish et al. (2009) EMBO J. 28:2387-99. The biochemical studies described in references 121 and 123 (in the years 2015 and 2016) confirm the early observations by Camenish et al. dating back to 2009.
- Line 372 and below lines 442-443: I strongly disagree with the statement that DDB2 and XPC display similar recognition mechanisms. DDB2 has a dedicated recognition pocket for the direct binding to dipyrimidine lesions. In contrast, XPC uses an indirect mode of recognition by binding to the undamaged strand while avoiding interactions with the dipyrimidine lesion. This difference should be clarified throughout the section on global-genome repair.
- The section 4.4 on damage verification (or "validation") is poorly supported by existing evidence. This section would be strengthened by discussing the biochemical experiments of Mathieu et al. (2010) Proc. Natl. Acad. Sci. USA 107:17545-50 demonstrating how the XPD helicase accomplishes this verification step. Also, in the follow-up paper (Curr. Biol. 23:204-12) Mathieu et al. identified two amino acid residues (Y192 and R196) that are responsible for the detection of UV lesions. I cannot see whether these papers are quoted (the publication list stops at No. 158) but in any case a more detailed discussion on XPD would help the reader to understand the critical scanning function of this DNA helicase.
Author Response
Reviewer 1
We would like to thank the reviewer for the constructive comments.
Lines 93-96: What is the presumed reason for the intriguing difference in the sequence preference of CPD formation between human and plant cells? Is there any hypothesis.
We introduced some hypotheses in the corresponding paragraph.
Line 265: What is the postulated role of PolII ubiquitination during the TCR process?
We better detailed the role of RNA Pol II ubiquitination and added new references.
Lines 265-268: The authors describe the push-forward action of CSB but not the possible backtracking function of TFIIH as described for example in the review by Lans et al. (2019) Nat. Rev. Mol. Cell. Biol. 20, 766-84.
We thank the reviewer for this comment. We rephrased the sentence and added this important contribution.
Line 337: The unique mechanism of substrate recognition by XPC protein (relying on the detection of a single-stranded configuration in the undamaged complementary sequence of the double helix) has first been reported by Maillard et al. (2007) PLoS Biol. 5:e79. This reviewer believes that credit should be given to this pioneering study that appeared before the description of a crystal structure of the yeast homolog.
We agree that this pioneering study must be cited in priority. We added this reference.
Lines 348-349: The mechanism by which XPC protein dynamically searches for DNA damage has been described in the context of living human cells by Camenish et al. (2009) EMBO J. 28:2387-99. The biochemical studies described in references 121 and 123 (in the years 2015 and 2016) confirm the early observations by Camenish et al. dating back to 2009.
We agree that this first study must be cited. We added this reference
Line 372 and below lines 442-443: I strongly disagree with the statement that DDB2 and XPC display similar recognition mechanisms. DDB2 has a dedicated recognition pocket for the direct binding to dipyrimidine lesions. In contrast, XPC uses an indirect mode of recognition by binding to the undamaged strand while avoiding interactions with the dipyrimidine lesion. This difference should be clarified throughout the section on global-genome repair.
In order to avoid any misunderstanding we rephrased the sentence according reviewer suggestion.
The section 4.4 on damage verification (or "validation") is poorly supported by existing evidence. This section would be strengthened by discussing the biochemical experiments of Mathieu et al. (2010) Proc. Natl. Acad. Sci. USA 107:17545-50 demonstrating how the XPD helicase accomplishes this verification step. Also, in the follow-up paper (Curr. Biol. 23:204-12) Mathieu et al. identified two amino acid residues (Y192 and R196) that are responsible for the detection of UV lesions. I cannot see whether these papers are quoted (the publication list stops at No. 158) but in any case a more detailed discussion on XPD would help the reader to understand the critical scanning function of this DNA helicase.
Indeed, we realized that references 158 to 176 have not been properly imported in the formatted version of the manuscript. We apologize for this unfortunate handling and we corrected this mistake.
The "Validation of NER recognition steps" paragraph has been rewritten to better discussed the different points raised by the reviewer.
Reviewer 2
We would like to thank the reviewer for the constructive comments.
The lack of a sharply defined focus. An effort is required to define the main scope of the review, harmonized throughout all parts of the manuscript (title, abstract, introduction, the rest the manuscript body and the final conclusions section). Should it be chiefly a survey of the mechanisms of formation of photodamage and of possible epigenetic determinants of damage generation? Or is the primary goal to review structural mechanisms of recognition of photodamage existing in different kingdoms of life and identify epigenetic components of repair? For instance, part 2 seams rather loosely bound to the rest of the manuscript and does not seem to be conceptually new. The authors may wish to reflect, whether this part is necessary.
We agree that the topic must be better defined. We devoted efforts to improve this point according reviewer suggestions.Uneven quality of writing. Sections 4.1 - 4.4 are very well structured and well written, whereas other sections (1; 2.1-2.2; 3.1-3.3) often lack precision, have problems with the use of terminology and contain some inadequate references. For instance, introduction suffers from too general statements and fails to present specific problems to be addressed. References 1, 2, 3, 4, 13, 26 and 27 are not optimal or wrong. Titles of subsections 3.1, 3.2 and 3.3 are imprecise or incorrect. Please correct these problems and carefully proofread the rest of the manuscript.
We thanks the reviewer for these constructive comments.
As suggested we choose new references and did our best to improve the introduction. We think that we better defined the problems addressed.
Part 4 reporting the repair pathways is informative and well written. I suggest to start with a brief overview of repair strategies used by different organisms (as in Figure 2), prior to in-depth description of the individual pathways.
We thank the reviewer for this suggestion. We added an introduction for this part of the review.
Conclusions (lines 472-496) do not really follow from the rest of the manuscript content. This section refers specifically to topics related to epigenetics and chromosomal organisation, which are covered by only a tiny proportion of the body of the literature reviewed in the manuscript. Apparently, there are problems with language and style in this section. For instance, I do not understand the statements on lines 473-475 and 480-481. The section contains references to publications not previously discussed in the manuscript.
We changed the shape and the content of this part.
Abstract could be improved. The last long sentence seems unnecessary. Sentences on lines 11-13 could be formulated more precisely; “review aims at providing an overview” is certainly not optimal (also on lines 39 and 472).
We improved the abstract.
As the terms “pre-recognition” and “recognition” steps of NER are often used in the text, it could be It could be helpful to use the same terminology in Figure 2.
We homogenized the terminology.
References 159 to 176 are not listed in the bibliography.
Indeed, we realized that references 158 to 176 have not been properly imported in the formatted version of the manuscript. We apologize for this unfortunate handling and we corrected this mistake.
Minor points:
All these points have been corrected/improved and highlighted in the revised version of the manuscript.
line 8: delete “wavelength”
line 10: “activated” seems to be a wrong word. Perhaps, “evolved”? See also lines 32 and 42
line 14: “ability to form photolesions” could be “susceptibility”
line 30: damage (singular) requires verb in singular form (“DNA damage is” or “DNA modifications are”). Please proofread the text for subject-verb agreement and use of singular/plural forms in general, e.g., lines 32, 57-58, 104 (change to “sequence”), 175 (change to “words”), 346, 350 (change to “different types of DNA damage”), 439 (change to “sites”.
Lines 40-42: expressions “genome reactivity” and “during the pathways” are not appropriate. Please re-phrase
line 45: DNA, not the genetic information, is a target of UV
line 50: “extend” should read “extent”, delete “even”
line 53: “identified” is probably a wrong word
line 57-59: please adjust the structure of the sentence. Please also revise sentences on lines lines 65-66, 91-93, 103-104, 154-156, 238-240, 333-336, 440-442
line 60: “pyramidic” should be “pyrimidine”
lines 61-62: “DNA helix distortions reflected by changes in DNA conformation” is an awkward expression
line 73: please use a dot (not circle) as a common symbol to indicate unpaired electron
line 164-165: “the neighboring proteome” must be a wrong expression
lines 177: “common ancestor of all living organisms expressed photolyase genes” is probably a wrong statement
lines 184-186: it may be useful to invert the order of the two sentences
line 238: delete “after the years of studies”
line 241: delete “basically”
line 258: delete “direct”
line 273: “conserved in eukaryotes”
line 277: probably should read “severe”
line 278: bacteria do not have pol II
line 309: should be “Phe334”
line 322: “is highlighted”
line 368: I suggest to change “Such mechanisms” to “The pre-recognition mechanism”
line 444: what are “mismatches related to UV-induced DNA damage”?
line 459: “single-strand DNA goes” should probably read “damaged DNA stand moves/is translocated”
Reviewer 2 Report
The manuscript by Johann to Berens and Molinier reviews the mechanisms of generation and repair of damage generated by UV. The authors address direct and indirect mechanisms of damage induced by UV in naked DNA and in chromatin, followed by overview of strategies used by organisms of different kingdoms of life to repair the major types of the UV photoadducts. This topic is important and the paper would be of potential interest for the IJMS readership.
This is a timely and comprehensive review. The mechanisms of damage recognition and repair are well described and adequately discussed; however, the rest of the manuscript requires a major revision. Several crucial sections of the manuscript, including conclusions and abstract need to be extensively edited or completely re-written to be considered for publication.
Major points:
1) The lack of a sharply defined focus. An effort is required to define the main scope of the review, harmonized throughout all parts of the manuscript (title, abstract, introduction, the rest the manuscript body and the final conclusions section). Should it be chiefly a survey of the mechanisms of formation of photodamage and of possible epigenetic determinants of damage generation? Or is the primary goal to review structural mechanisms of recognition of photodamage existing in different kingdoms of life and identify epigenetic components of repair? For instance, part 2 seams rather loosely bound to the rest of the manuscript and does not seem to be conceptually new. The authors may wish to reflect, whether this part is necessary.
2) Uneven quality of writing. Sections 4.1 - 4.4 are very well structured and well written, whereas other sections (1; 2.1-2.2; 3.1-3.3) often lack precision, have problems with the use of terminology and contain some inadequate references. For instance, introduction suffers from too general statements and fails to present specific problems to be addressed. References 1, 2, 3, 4, 13, 26 and 27 are not optimal or wrong. Titles of subsections 3.1, 3.2 and 3.3 are imprecise or incorrect. Please correct these problems and carefully proofread the rest of the manuscript.
3) Part 4 reporting the repair pathways is informative and well written. I suggest to start with a brief overview of repair strategies used by different organisms (as in Figure 2), prior to in-depth description of the individual pathways.
4) Conclusions (lines 472-496) do not really follow from the rest of the manuscript content. This section refers specifically to topics related to epigenetics and chromosomal organisation, which are covered by only a tiny proportion of the body of the literature reviewed in the manuscript. Apparently, there are problems with language and style in this section. For instance, I do not understand the statements on lines 473-475 and 480-481. The section contains references to publications not previously discussed in the manuscript.
5) Abstract could be improved. The last long sentence seems unnecessary. Sentences on lines 11-13 could be formulated more precisely; “review aims at providing an overview” is certainly not optimal (also on lines 39 and 472)
6) As the terms “pre-recognition” and “recognition” steps of NER are often used in the text, it could be It could be helpful to use the same terminology in Figure 2.
7) References 159 to 176 are not listed in the bibliography.
Minor points:
line 8: delete “wavelength”
line 10: “activated” seems to be a wrong word. Perhaps, “evolved”? See also lines 32 and 42
line 14: “ability to form photolesions” could be “susceptibility”
line 30: damage (singular) requires verb in singular form (“DNA damage is” or “DNA modifications are”). Please proofread the text for subject-verb agreement and use of singular/plural forms in general, e.g., lines 32, 57-58, 104 (change to “sequence”), 175 (change to “words”), 346, 350 (change to “different types of DNA damage”), 439 (change to “sites”.
Lines 40-42: expressions “genome reactivity” and “during the pathways” are not appropriate. Please re-phrase
line 45: DNA, not the genetic information, is a target of UV
line 50: “extend” should read “extent”, delete “even”
line 53: “identified” is probably a wrong word
line 57-59: please adjust the structure of the sentence. Please also revise sentences on lines lines 65-66, 91-93, 103-104, 154-156, 238-240, 333-336, 440-442
line 60: “pyramidic” should be “pyrimidine”
lines 61-62: “DNA helix distortions reflected by changes in DNA conformation” is an awkward expression
line 73: please use a dot (not circle) as a common symbol to indicate unpaired electron
line 164-165: “the neighboring proteome” must be a wrong expression
lines 177: “common ancestor of all living organisms expressed photolyase genes” is probably a wrong statement
lines 184-186: it may be useful to invert the order of the two sentences
line 238: delete “after the years of studies”
line 241: delete “basically”
line 258: delete “direct”
line 273: “conserved in eukaryotes”
line 277: probably should read “severe”
line 278: bacteria do not have pol II
line 309: should be “Phe334”
line 322: “is highlighted”
line 368: I suggest to change “Such mechanisms” to “The pre-recognition mechanism”
line 444: what are “mismatches related to UV-induced DNA damage”?
line 459: “single-strand DNA goes” should probably read “damaged DNA stand moves/is translocated”
Author Response
Reviewer 2
We would like to thank the reviewer for the constructive comments.
The lack of a sharply defined focus. An effort is required to define the main scope of the review, harmonized throughout all parts of the manuscript (title, abstract, introduction, the rest the manuscript body and the final conclusions section). Should it be chiefly a survey of the mechanisms of formation of photodamage and of possible epigenetic determinants of damage generation? Or is the primary goal to review structural mechanisms of recognition of photodamage existing in different kingdoms of life and identify epigenetic components of repair? For instance, part 2 seams rather loosely bound to the rest of the manuscript and does not seem to be conceptually new. The authors may wish to reflect, whether this part is necessary.
We agree that the topic must be better defined. We devoted efforts to improve this point according reviewer suggestions.
Uneven quality of writing. Sections 4.1 - 4.4 are very well structured and well written, whereas other sections (1; 2.1-2.2; 3.1-3.3) often lack precision, have problems with the use of terminology and contain some inadequate references. For instance, introduction suffers from too general statements and fails to present specific problems to be addressed. References 1, 2, 3, 4, 13, 26 and 27 are not optimal or wrong. Titles of subsections 3.1, 3.2 and 3.3 are imprecise or incorrect. Please correct these problems and carefully proofread the rest of the manuscript.
We thanks the reviewer for these constructive comments.
As suggested we choose new references and did our best to improve the introduction. We think that we better defined the problems addressed.
Part 4 reporting the repair pathways is informative and well written. I suggest to start with a brief overview of repair strategies used by different organisms (as in Figure 2), prior to in-depth description of the individual pathways.
We thank the reviewer for this suggestion. We added an introduction for this part of the review.
Conclusions (lines 472-496) do not really follow from the rest of the manuscript content. This section refers specifically to topics related to epigenetics and chromosomal organisation, which are covered by only a tiny proportion of the body of the literature reviewed in the manuscript. Apparently, there are problems with language and style in this section. For instance, I do not understand the statements on lines 473-475 and 480-481. The section contains references to publications not previously discussed in the manuscript.
We changed the shape and the content of this part.
Abstract could be improved. The last long sentence seems unnecessary. Sentences on lines 11-13 could be formulated more precisely; “review aims at providing an overview” is certainly not optimal (also on lines 39 and 472).
We improved the abstract.
As the terms “pre-recognition” and “recognition” steps of NER are often used in the text, it could be It could be helpful to use the same terminology in Figure 2.
We homogenized the terminology.
References 159 to 176 are not listed in the bibliography.
Indeed, we realized that references 158 to 176 have not been properly imported in the formatted version of the manuscript. We apologize for this unfortunate handling and we corrected this mistake.
Minor points:
All these points have been corrected/improved and highlighted in the revised version of the manuscript.
line 8: delete “wavelength”
line 10: “activated” seems to be a wrong word. Perhaps, “evolved”? See also lines 32 and 42
line 14: “ability to form photolesions” could be “susceptibility”
line 30: damage (singular) requires verb in singular form (“DNA damage is” or “DNA modifications are”). Please proofread the text for subject-verb agreement and use of singular/plural forms in general, e.g., lines 32, 57-58, 104 (change to “sequence”), 175 (change to “words”), 346, 350 (change to “different types of DNA damage”), 439 (change to “sites”.
Lines 40-42: expressions “genome reactivity” and “during the pathways” are not appropriate. Please re-phrase
line 45: DNA, not the genetic information, is a target of UV
line 50: “extend” should read “extent”, delete “even”
line 53: “identified” is probably a wrong word
line 57-59: please adjust the structure of the sentence. Please also revise sentences on lines lines 65-66, 91-93, 103-104, 154-156, 238-240, 333-336, 440-442
line 60: “pyramidic” should be “pyrimidine”
lines 61-62: “DNA helix distortions reflected by changes in DNA conformation” is an awkward expression
line 73: please use a dot (not circle) as a common symbol to indicate unpaired electron
line 164-165: “the neighboring proteome” must be a wrong expression
lines 177: “common ancestor of all living organisms expressed photolyase genes” is probably a wrong statement
lines 184-186: it may be useful to invert the order of the two sentences
line 238: delete “after the years of studies”
line 241: delete “basically”
line 258: delete “direct”
line 273: “conserved in eukaryotes”
line 277: probably should read “severe”
line 278: bacteria do not have pol II
line 309: should be “Phe334”
line 322: “is highlighted”
line 368: I suggest to change “Such mechanisms” to “The pre-recognition mechanism”
line 444: what are “mismatches related to UV-induced DNA damage”?
line 459: “single-strand DNA goes” should probably read “damaged DNA stand moves/is translocated”

Round 2
Reviewer 1 Report
With the revisions, the review is now ready to be published.
Author Response
We thank the reviewer
Reviewer 2 Report
The authors have revised the manuscript along the previous criticism points to improve the contents and the structure of the paper. Many parts are still unnecessarily difficult to understand (for example fragments/sentences on lines 48-535, 42-543, 552-554). Some other points for editing:
line 16: "among the different DNA repair pathways" is not a right expression. I may suggest through the different pathways? across the pathways?
lines 37-38: "these wavelengths can lead to deleterious effects". I suggest "these types of irradiation"
line 151: The term "damageability" (used several times in the text) is not usual; "damageability to UV-R" is probably not correct.
line 556: "environmental cues" is probably a wrong expression
Author Response
We thank the reviewer for these suggestions. We highlighted in blue the changes
Many parts are still unnecessarily difficult to understand (for example fragments/sentences on lines 48-535, 42-543, 552-554).
We rephrased these sentences.
line 16: "among the different DNA repair pathways" is not a right expression. I may suggest through the different pathways? across the pathways?
lines 37-38: "these wavelengths can lead to deleterious effects". I suggest "these types of irradiation"
We changed these sentences according reviewer's suggestions.
line 151: The term "damageability" (used several times in the text) is not usual; "damageability to UV-R" is probably not correct.
We removed this word from the manuscript and used another wording.
line 556: "environmental cues" is probably a wrong expression
We rephrased this sentence.